# Regression-tree Tuning in a Streaming Setting

**Samory Kpotufe**[*]
Toyota Technological Institute at Chicago[†]
firstname@ttic.edu

**Francesco Orabona**[*]
Toyota Technological Institute at Chicago
francesco@orabona.com

## Abstract

We consider the problem of maintaining the data-structures of a partition-based regression procedure in a setting where the training data arrives sequentially over time. We prove that it is possible to maintain such a structure in time $\mathcal{O}\left(\log n\right)$ at any time step $n$ while achieving a nearly-optimal regression rate of $\tilde{\mathcal{O}}\left(n^{-2/(2+d)}\right)$ in terms of the unknown metric dimension $d$. Finally we prove a new regression lower-bound which is independent of a given data size, and hence is more appropriate for the streaming setting.

## 1 Introduction

Traditional nonparametric regression such as kernel or $k$-NN can be expensive to estimate given modern large training data sizes. It is therefore common to resort to cheaper methods such as tree-based regression which precompute the regression estimates over a partition of the data space [7]. Given a future query $x$, the estimate $f_n(x)$ simply consists of finding the *closest* cell of the partition by traversing an appropriate tree-structure and returning the precomputed estimate. The partition and precomputed estimates depend on the training data and are usually maintained in batch-mode. We are interested in maintaining such a partition and estimates in a real-world setting where the training data arrives sequentially over time. Our constraints are that of fast-update at every time step, while maintaining a near-minimax regression error-rate at any point in time.

The error-rate of tree-based regression is well known to depend on the *size* of the partition's cells. We will call this size the *binwidth*. The minimax-optimal binwidth $\epsilon_n$ is known to be of the form $\mathcal{O}\left(n^{-1/(2+d)}\right)$, assuming a training data of size $n$ from a metric space of unknown dimension $d$, and unknown Lipschitz target function $f$. This setting of $\epsilon_n$ would then yield a minimax error rate of $\mathcal{O}\left(n^{-2/(2+d)}\right)$. Thus, the dimension $d$ is the most important problem variable entering the rate (and the tuning of $\epsilon_n$), while other problem variables such as the Lipschitz properties of $f$ are less crucial in comparison. The main focus of this work is therefore that of adapting to the unknown $d$ while maintaining fast partition estimates in a streaming setting.

A first idea would be to start with an initial dimension estimation phase (where the regression estimates are suboptimal), and using the estimated dimension for subsequent data in a following phase, which leaves only the problem of maintaining partition estimates over time. However, while this sounds reasonable, it is generally unclear when to confidently stop such an initial phase since this would depend on the unknown $d$ and the distribution of the data.

Our solution is to interleave dimension estimation with regression updates as the data arrives sequentially. However the algorithm never relies on the estimated dimensions and views them rather as *guesses* $d_i$. Even if $d_i \neq d$, it is kept as long as it is not hurtful to regression performance. The guess $d_i$ is discarded once we detect that it hurts the regression, a new $d_{i+1}$ is then estimated and a new phase $i+1$ is started. The decision to discard $d_i$ relies on monitoring quantities that play into the tradeoff between regression variance and bias, more precisely we monitor the size of the partition

---

[*]SK and FO contributed equally to this paper.

[†]Other affiliation: Max Planck Institute for Intelligent Systems, Germany

and the partition's binwidth $\epsilon_n$. We note that the idea can be applied to other forms of regression where other quantities control the regression variance and bias (see longer version of the paper).

## 1.1 Technical Overview of Results

We assume that training data $(x_i, Y_i)$ is sampled sequentially over time, $x_i$ belongs to a general metric space $\mathcal{X}$ of unknown dimension $d$, and $Y_i$ is real. The exact setup is given in Section 2.

The algorithm (presented in Section 2.3) maintains regression estimates for all training samples $x^n \triangleq \{x_t\}_{t=1}^n$ arriving over time, while constantly updating a partition of the data and partition binwidth. At any time $t = n$, all updates are proveably of order $\log n$ with constants depending on the unknown dimension $d$ of $\mathcal{X}$.

At time $t = n$, the estimate for a query point $x$ is given by the precomputed estimate for the closest point to $x$ in $x^n$, which can be found fast using an off-the-shelf similarity search structure, such as those of [2, 10]. We prove that the $L_2$ error of the algorithm is $\tilde{\mathcal{O}}\left(n^{-2/(2+d)}\right)$, nearly optimal in terms of the unknown dimension $d$ of the metric $\mathcal{X}$.

Finally, we prove a new lower-bound for regression on a generic metric $\mathcal{X}$, where the worst-case distribution is the same as $n$ increases. Note that traditional lower-bounds for the offline setting derive a different worst-case distribution for each sample size $n$. Thus, our lower-bound is more appropriate to the streaming setting where the data arrives over time from the same distribution.

The results are discussed in more technical details in Section 3.

## 1.2 Related Work

Although various interesting heuristics have been proposed for maintaining tree-based learners in streaming settings (see e.g. [1, 5, 11, 15]), the problem has not received much theoretical attention. This is however an important problem given the growing size of modern datasets, and given that in many modern applications, training data is actually acquired sequentially over time and incremental updates have to be efficient (see e.g. Robotics [12, 16], Finance [8]).

The most closely related theoretical work is that of [6] which treats the problem of tuning a local polynomial regressor where the training data is acquired over time. Their setting however is that of a Euclidean space where $d$ is known (ambient Euclidean dimension). [6] is thus concerned with maintaining a minimax error rate w.r.t. the known dimension $d$, while efficiently tuning regression bandwidth.

A possible alternative to the method analyzed here is to employ some form of cross-validation or even online solutions based on *mixture of experts* [3], by keeping track of different partitions, each corresponding to some setting of the bindwidth $\epsilon_n$. This is however likely expensive to maintain in practice if good prediction performance is desired.

## 2 Preliminaries

### 2.1 Notions of metric dimension

We consider the following notion of dimension which extends traditional notions of dimension (e.g. Euclidean dimension and manifold dimension) to general metric spaces [4]. We assume throughout, w.l.o.g. that the space $\mathcal{X}$ has diameter at most 1 under a metric $\rho$.

**Definition 1.** *The metric measure space $(\mathcal{X}, \mu, \rho)$ has* **metric measure-dimension** *$d$, if there exist $\check{C}_\mu, \hat{C}_\mu$ such that for all $\epsilon > 0$, and $x \in \mathcal{X}$, $\check{C}_\mu \epsilon^d \leq \mu(B(x, \epsilon)) \leq \hat{C}_\mu \epsilon^d$.*

The assumption of finite metric-measure dimension ensures that the measure $\mu$ has mass everywhere on the space $\rho$. This assumption is a generalization (to a metric space) of common assumptions where the measure has an upper and lower-bounded density on a compact Euclidean space, however is more general in that it does not require the measure $\mu$ to have a density (relative to any reference measure). The metric-measure dimension implies the following other notion of metric dimension.

**Definition 2.** *The metric $(\mathcal{X}, \rho)$ has* **metric dimension** *$d$, if there exists $\hat{C}_\rho$ such that, for all $\epsilon > 0$, $\mathcal{X}$ has an $\epsilon$-cover of size at most $\hat{C}_\rho \epsilon^{-d}$.*

The relation between the two notions of dimension is stated in the following lemma of [9], which allows us to use either notion as needed.

**Lemma 1** ([9]). *If $(\mathcal{X}, \mu, \rho)$ has metric-measure dimension d, then there exists $\check{C}_\rho, \hat{C}_\rho$ such that, for all $\epsilon > 0$, any ball $B(x, r)$ centered on $(\mathcal{X}, \rho)$ has an $\epsilon r$-cover of size in $[\check{C}_\rho \epsilon^{-d}, \hat{C}_\rho \epsilon^{-d}]$.*

### 2.2 Problem Setup

We receive data pairs $(x_1, Y_1), (x_2, Y_2), \ldots$ sequentially, i.i.d. The input $x_t$ belongs to a metric measure space $(\mathcal{X}, \rho, \mu)$ of diameter at most 1, and of metric measure dimension d. The output $Y_t$ belongs to a subset of $\mathbb{R}$ of bounded diameter $\Delta_Y$, and satisfies $Y_t = f(x_t) + \eta(x_t)$. The noise $\eta(x_t)$ has 0 mean. The unknown function $f$ is assumed to be $\lambda$-Lipschitz w.r.t. $\rho$ for an unknown parameter $\lambda$, that is $\forall x, x' \in \mathcal{X}, \quad |f(x) - f(x')| \leq \lambda \rho(x, x')$.

$L_2$ **error:** Our main performance result bounds the excess $L_2$ risk

$$\mathop{\mathbb{E}}_{x^n, Y^n} \|f_n - f\|_{2,\mu}^2 \triangleq \mathop{\mathbb{E}}_{x^n, Y^n} \mathop{\mathbb{E}}_{X} |f_n(X) - f(X)|^2 .$$

We will often also be interested in the average error on the training sample: recall that at any time $t$, an estimate $f_t(x_s)$ of $f$ is produced for every $x_s \in x^t$. The average error on $x^n$ at $t = n$ is denoted

$$\mathbb{E}_n \mathop{\mathbb{E}}_{Y^n} |f_n(X) - f(X)|^2 \triangleq \frac{1}{n} \sum_{t=1}^{n} |f_n(x_t) - f(x_t)|^2 .$$

### 2.3 Algorithm

The procedure (Algorithm 1) works by partitioning the data into small regions of size roughly $\epsilon_t/2$ at any time $t$, and computing the regression estimate of the centers of each region. All points falling in the same region (identified by a center point) are assigned the same regression estimate: the average $Y$ values of all points in the region.

The procedure works in phases, where each Phase $i$ corresponds to a guess $d_i$ of the metric dimension $d$. Where $\epsilon_t$ might have been set to $t^{-1/(2+d)}$ if we knew $d$, we set it to $t_i^{-1/(2+d_i)}$ where $t_i$ is the current time step within Phase $i$.

We ensure that in each phase our guess $d_i$ does not hurt the variance-bias tradeoff of the estimates: this is done by monitoring the size of the partition ($|\mathbf{X}_i|$ in the algorithm), which controls the variance (see analysis in Section 4), relative to the bindwidth $\epsilon_t$, which controls bias. Whenever $|\mathbf{X}_i|$ is too large relative to $\epsilon_t$, the variance of the procedure is likely too large, so we start a new phase with an new guess of $d_i$.

Since the algorithm maintains at any time $n$ an estimate $f_n(x_t)$ for all $x_t \in x^n$, for any query point $x \in \mathcal{X}$, we simply return $f_n(x) = f_n(x_t)$ where $x_t$ is the closest point to $x$ in $x^n$.

Despite having to adaptively tune to the unknown $d$, the main computation at each time step consists of just a 2-approximate nearest neighbor search for the closest center. These searches can be done fast in time $\mathcal{O}(\log n)$ by employing the off-the-shelf online search-procedure of [10]. This is emphasized in Lemma 2 below.

Finally, the algorithm employs a constant $\bar{C}$ which is assumed to upper-bound the constant $C_\rho$ in Definition 2. This is a minor assumption since $C_\rho$ is generally taken to be small, e.g. 1, in machine learning literature, and is exactly quantifieable for various metrics [4, 10].

## 3 Discussion of Results

### 3.1 Time complexity

The time complexity of updates is emphasized in the following Lemma.

**Lemma 2.** *Suppose $(\mathcal{X}, \rho, \mu)$ has metric dimension d. Then there exist $C$ depending on d such that all computations of the algorithm at any time $t = n$ can be done in time $C \log n$.*

**Algorithm 1** Incremental tree-based regressor.

1: **Initialize:** $i = 1$, $d_i = 1$, $t_i = 0$, Centers $\mathbf{X}_i = \emptyset$
2: **for** $t = 1, 2, \ldots, T$ **do**
3:      Receive $(x_t, y_t)$
4:      $t_i \leftarrow t_i + 1$ // counts the time steps within Phase $i$
5:      $\epsilon_t \leftarrow t_i^{-1/(2+d_i)}$
6:      Set $x_s \in \mathbf{X}_i$ to the 2-approximate nearest neighbor of $x_t$
7:      **if** $\rho(x_t, x_s) \leq \epsilon_t$ **then**
8:          Assign $x_t$ to $x_s$
9:          $f_n(x_s) \leftarrow$ update average $Y$ for center $x_s$ with $y_t$
10:         For every $r \leq t$ assigned to $x_s$, $f_n(x_r) = f_n(x_s)$
11:      **else**
12:         **if** $|\mathbf{X}_i| + 1 > \hat{C} \, 4^{d_i} \epsilon_t^{-d_i}$ **then**
13:             // Start of Phase $i + 1$
14:             $d_{i+1} \leftarrow \left\lceil \log(\frac{|\mathbf{X}_i|+1}{\hat{C}}) / \log(4/\epsilon_t) \right\rceil$
15:             $i \leftarrow i + 1$
16:         **end if**
17:         Add $x_t$ as a new center in $\mathbf{X}_i$
18:      **end if**
19: **end for**

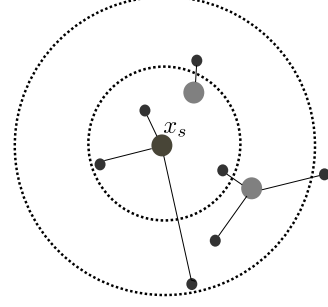

Figure 1: As $\epsilon_t$ varies over time, a ball around a center $x_s$ can eventually contain both points assigned to $x_s$ and points non-assigned to it, and even contain other centers. This results in a complex partitioning of the data.

*Proof.* The main computation at time $n$ consists of finding the 2-approximate nearest neighbor $x_n$ in $\mathbf{X}_i$ and update the data structure for the nearest neighbor search. These centers are all at least $\frac{\epsilon_n}{2}$ far-apart. Using the results of [10], this can be done online in time $\mathcal{O}\left(\log(1/\epsilon_n) + \log\log(1/\epsilon_n)\right)$. $\square$

## 3.2 Convergence rates

The main theorem below bounds the $L_2$ error of the algorithm at any given point in time. The main difficulty is in the fact that the data is partitioned in a complicated way due to the ever-changing bindwidth $\epsilon_t$: every ball around a center can eventually contain both points assigned to the center and points not assigned to the center, in fact can contain other centers (see Figure 1). This makes it hard to get a handle on the number of points assigned to a single center $x_t$ (contributing to the variance of $f_n(x_t)$) and the distance between points assigned to the same center (contributing to the bias). This is not the case in classical analyses of tree-based regression since the data partitioning is usually clearly defined.

The problem is handled by first looking at the average error over points in $x^n$, which is less difficult.

**Theorem 1.** *Suppose the space $(\mathcal{X}, \mu, \rho)$ has metric-measure dimension $d$.*

*For any $x \in \mathcal{X}$, define $f_n(x) = f_n(x_t)$ where $x_t$ is the closest point to $x$ in $x^n$. Then at any time $t = n$, we have for some $C$ independent of $n$,*

$$\mathbb{E}_{x^n, Y^n} \|f_n - f\|_{2,\mu}^2 \leq C(d \log n) \sup_{x^n} \mathbb{E}_n \mathbb{E}_{Y^n} \|f_n(X) - f(X)\|^2 + C\lambda^2 \left(\frac{d \log n}{n}\right)^{2/d} + \frac{\Delta_Y^2}{n} \, .$$

*If the algorithm parameter $\hat{C} \geq \hat{C}_\rho$, then for some constant $C'$ independent of $n$, we have at any time $n$ that*

$$\sup_{x^n} \mathbb{E}_n \mathbb{E}_{Y^n} |f_n(X) - f(X)|^2 \leq C' \left(\Delta_Y^2 + \lambda^2\right) n^{-2/(2+d)} \, .$$

The convergence rate is therefore $\tilde{\mathcal{O}}(n^{-2/(2+d)})$, nearly optimal in terms of the unknown $d$ (up to a $\log n$ factor). In the simulation of Figure 2(Left) we compare our procedure to tree-based regressors with a fixed setting of $d$ and of $\epsilon_t = t^{-1/(2+d)}$. We use the classic rotating-Teapot dataset, where the target output values are the cosine of the rotation angles. Our method attains the same performance as the one with the right fixed setting of $d$.

As alluded to above, the proof of Theorem 1 proceeds by first bounding the average error $\mathbb{E}_n \mathbb{E}_{Y^n} |f_n(X) - f(X)|^2$ on the sample $x^n$. Interestingly, the analysis of the average error is

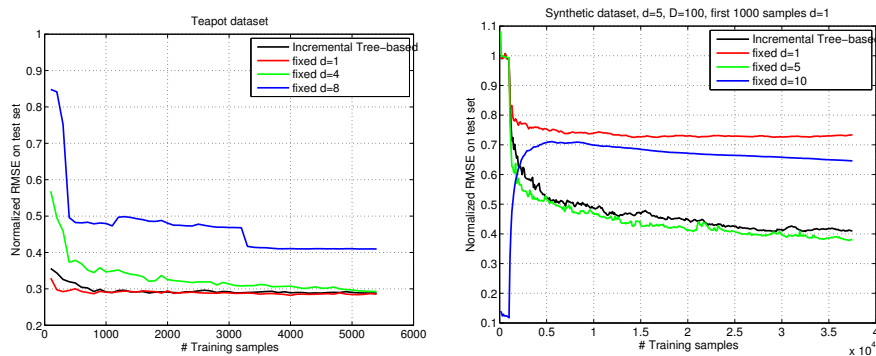

Figure 2: Simulation results on Teapot (Left) and Synthetic (Right) datasets. $\hat{C}$ set to 1, size of the test sets 1800 and 12500, respectively.

of a worst-case nature where the data $x_1, x_2, \ldots$ is allowed to arrive adversarially (see analysis of Section 4.1). This shows a sense in which the algorithm is robust to bad dimension estimates: the average error is of the optimal form in terms of $d$, even though the data could trick us into picking a bad guess $d_i$ of $d$. Thus the insights behind the algorithm are perhaps of wider applicability to problems of a more adversarial nature. This is shown empirically in Figure 2(Right), where we created a synthetic datasets with $d = 5$, while the first 1000 samples are from a line in $\mathcal{X}$. An algorithm that estimates the dimension in a first phase would end up using the suboptimal setting $d = 1$, while our algorithm robustly updates its estimate over time.

As mentioned in the introduction, the same insights can be applied to other forms of regression in a streaming setting. We show in the longer version of the paper a procedure more akin to kernel regression, which employs other quantities (appropriate to the method) to control the bias-variance tradeoff while deciding on keeping or rejecting the guess $d_i$.

### 3.3 Lower-bounds

We have to produce a distribution for which the problem is hard, and which matches our streaming setting as well as possible. With this in mind, our lower-bound result differs from existing non-parametric lower-bounds by combining two important aspects. First, the lower-bound holds for any given metric measure space $(\mathcal{X}, \rho, \mu)$ with finite measure-dimension: we constrain the worst-case distribution to have the marginal $\mu$ that nature happens to choose. In contrast, lower-bounds in literature would commonly pick a suitable marginal on the space $\mathcal{X}$ [13, 14]. Second, the worst-case distribution does not depend on the sample size as is common in literature. Instead, we show that the rate of $n^{-2/(2+d)}$ holds for infinitely many $n$ for a distribution fixed beforehand. This is more appropriate for the online setting where the data is generated over time from a fixed distribution.

The lower-bound result of [9] also holds for a given measure space $(\mathcal{X}, \mu, \rho)$, but the worst-case distribution depends on sample size. A lower-bound of [7] holds for infinitely many $n$, but is restricted to distributions on a Euclidean cube, and thus benefits from the regularity of the cube. Our result combines some technical intuition from these two results in a way described in Section 4.3.

We need the following definition.

**Definition 3.** *Given a metric-measure space $(\mathcal{X}, \mu, \rho)$, we let $\mathcal{D}_{\mu,\lambda}$ denote the set of distributions on $X, Y, X \in \mathcal{X}, Y \in \mathbb{R}$, where the marginal on $\mathcal{X}$ is $\mu$, and where the function $f(x) = \mathbb{E}[Y|X = x]$ is $\lambda$-Lipschitz w.r.t. $\rho$.*

**Theorem 2.** *Let $(\mathcal{X}, \mu, \rho)$ be a metric space of diameter $1$, and metric-measure dimension $d$. For any $n \in \mathbb{N}$, define $r_n^2 = (\lambda^2 n)^{-2/(2+d)}$. Pick any positive sequence $\{\beta_n\}_{n \in \mathbb{N}}$, $\beta_n = o\left(\lambda^2 r_n^2\right)$. There exists an indexing subsequence $\{n_t\}_{t \in \mathbb{N}}$, $n_{t+1} > n_t$, such that*

$$\inf_{\{f_n\}} \sup_{\mathcal{D}_{\mu,\lambda}} \lim_{t \to \infty} \frac{\mathbb{E}_{X^{n_t}, Y^{n_t}} \|f_{n_t} - f\|_{2,\mu}^2}{\beta_{n_t}} = \infty,$$

*where the infimum is taken over all sequences $\{f_n\}$ of estimators $f_n : X^n, Y^n \mapsto L_{2,\mu}$.*

By the statement of the theorem, if we pick any rate $\beta_n$ faster than $n^{-2/(2+d)}$, then there exists a distribution with marginal $\mu$ for which $\mathbb{E} \left\| f_n - f \right\|^2 / \beta_n$ either diverges or tends to $\infty$.

## 4   Analysis

We first analyze the average error of the algorithm over the data $x^n$ in Section 4.1. The proof of the main theorem follows in Section 4.2.

### 4.1   Bounds on Average Error

We start by bounding the average error on the sample $x^n$ at time $n$, that is we upper-bound $\mathbb{E}_n \, \mathbb{E}_{Y^n} \left| f_n(X) - f(X) \right|^2$.

The proof idea of the upper bound is the following. We bound the error in a given phase (Lemma 4), then combine these errors over all phases to obtain the final bounds (Corollary 1). To bound the error in a phase, we decompose the error in terms of squared bias and variance. The main technical difficulty is that the bandwidth $\epsilon_t$ varies over time and thus points at varying distances are included in each estimate. Nevertheless, if $n_i$ is the number of steps in Phase $i$, we will see that both average squared bias and variance can be bounded by roughly $n_i^{-2/(2+d_i)}$.

Finally, the algorithm ensures that the guess $d_i$ is always an under-estimate of the unknown dimension $d$, as proven in Lemma 3 (proof in the supplemental appendix), so integrating over all phases yields an adaptive bound on the average error. We assume throughout this section that the space $(\mathcal{X}, \rho)$ has dimension $d$ for some $\hat{C}_\rho$ (see Def. 2).

**Lemma 3.** *Suppose the algorithm parameter $\hat{C} \geq \hat{C}_\rho$. The following invariants hold throughout the procedure for all phases $i \geq 1$ of Algorithm 1:*

- *$i \leq d_i \leq d$.*

- *For any $t \in$ Phase $i$ we have $|\mathbf{X}_i| \leq \hat{C} \, 4^{d_i} \epsilon_t^{-d_i}$.*

**Lemma 4** (Bound on single phase)**.** *Suppose the algorithm parameter $\hat{C} \geq \hat{C}_\rho$. Consider Phase $i \geq 1$ and suppose this phase lasts $n_i$ steps. Let $\mathbb{E}_{n_i}$ denote expectation relative to the uniform choice of $X$ out of $\{x_t : t \in$ Phase $i\}$. We have the following bound:*

$$\underset{n_i}{\mathbb{E}} \, \underset{Y^n}{\mathbb{E}} \, \left| f_n(X) - f(X) \right|^2 \leq \left( \hat{C} 4^d \Delta_Y^2 + 12\lambda^2 \right) n_i^{-\frac{2}{2+d}}.$$

*Proof.* Let $\mathbf{X}_i(X)$ denote the center closest to $X$ in $\mathbf{X}_i$. Suppose $\mathbf{X}_i(X) = x_s, s \in [n]$, we let $n_{x_s}$ denote the number of points assigned to the center $x_s$. We use the notation $x_t \to x_s$ to say that $x_t$ is assigned to center $x_s$.

First fix $X \in \{x_t : t \in$ Phase $i\}$ and let $x_s = \mathbf{X}_i(x_t)$. Define $\tilde{f}_n(X) \equiv \mathbb{E}_{Y^n} f_n(X) = \frac{1}{n_{x_s}} \sum_{x_t \to x_s} f(x_t)$. We proceed with the following standard bias-variance decomposition

$$\underset{Y^n}{\mathbb{E}} \left| f_n(X) - f(X) \right|^2 = \underset{Y^n}{\mathbb{E}} \left| f_n(X) - \tilde{f}_n(X) \right|^2 + \left| \tilde{f}_n(X) - f(X) \right|^2. \tag{1}$$

Let $X = x_r, r \geq s$. We first bound the bias term. Using the Lipschitz property of $f$, and Jensen's inequality, we have

$$\left| \tilde{f}_n(X) - f(X) \right|^2 \leq \left( \frac{1}{n_{x_s}} \sum_{x_t \to x_s} \lambda \rho \left( x_r, x_t \right) \right)^2 \leq \frac{1}{n_{x_s}} \sum_{x_t \to x_s} \lambda^2 \rho \left( x_r, x_t \right)^2$$

$$\leq \frac{2\lambda^2}{n_{x_s}} \sum_{x_t \to x_s} \left( \rho \left( x_r, x_s \right)^2 + \rho \left( x_s, x_t \right)^2 \right) \leq \frac{2\lambda^2}{n_{x_s}} \sum_{x_t \to x_s} \left( \epsilon_r^2 + \epsilon_t^2 \right).$$

The variance term is easily bounded as follows:

$$\underset{Y^n}{\mathbb{E}} \left| f_n(X) - \tilde{f}_n(X) \right|^2 = \sum_{x_t \to x_s} \frac{\mathbb{E}_{Y^n} \left| Y_t - f(x_t) \right|^2}{n_{x_s}^2} \leq \frac{\Delta_Y^2}{n_{x_s}}.$$

Now take the expectation over $X \sim \mathcal{U}\{x_t : t \in \text{Phase } i\}$. We have:

$$\mathbb{E}_{n_i} \mathbb{E}_{Y^n} |f_n(X) - f(X)|^2 = \sum_{x_s \in \mathbf{X}_i} \mathbb{E}_{n} \mathbb{E}_{Y^n} |f_n(X) - f(X)|^2 \cdot \mathbf{1}\{X \to x_s\}$$

$$\leq \frac{1}{n_i} \sum_{x_s \in \mathbf{X}_i} \sum_{x_r \to x_s} \left( \frac{\Delta_Y^2}{n_{x_s}} + \frac{2\lambda^2}{n_{x_s}} \sum_{x_t \to x_s} (\epsilon_r^2 + \epsilon_t^2) \right)$$

$$= \frac{1}{n_i} \sum_{x_s \in \mathbf{X}_i} \Delta_Y^2 + \frac{2\lambda^2}{n_i} \sum_{x_s \in \mathbf{X}_i} \frac{1}{n_{x_s}} \sum_{x_r \to x_s} \sum_{x_t \to x_s} (\epsilon_r^2 + \epsilon_t^2)$$

$$= \frac{\Delta_Y^2 \cdot |\mathbf{X}_i|}{n_i} + \frac{4\lambda^2}{n_i} \sum_{x_s \in \mathbf{X}_i} \sum_{x_t \to x_s} \epsilon_t^2 = \frac{\Delta_Y^2 \cdot |\mathbf{X}_i|}{n_i} + \frac{4\lambda^2}{n_i} \sum_{t \in \text{Phase } i} \epsilon_t^2 .$$

To bound the last term, we have

$$\sum_{t \in \text{Phase } i} \epsilon_t^2 = \sum_{t_i \in [n_i]} t^{-\frac{2}{2+d_i}} \leq \int_0^{n_i} \tau^{-\frac{2}{2+d_i}} \, d\tau \leq 3 n_i^{1 - \frac{2}{2+d_i}} .$$

Combine with the previous derivation and with both statements of Lemma 3 to get

$$\mathbb{E}_{n_i} \mathbb{E}_{Y^n} |f_n(X) - f(X)|^2 \leq \frac{\Delta_Y^2 \cdot |\mathbf{X}_i|}{n_i} + 12\lambda^2 n_i^{-\frac{2}{2+d_i}} \leq \left( \hat{C} \, 4^d \, \Delta_Y^2 + 12\lambda^2 \right) n_i^{-\frac{2}{2+d}} .$$

$\square$

**Corollary 1** (Combined phases). *Suppose the algorithm parameter $\hat{C} \geq \hat{C}_\rho$, then we have*

$$\mathbb{E}_{n} \mathbb{E}_{Y^n} |f_n(X) - f(X)|^2 \leq 2 \left( \hat{C} \, 4^d \Delta_Y^2 + 12\lambda^2 \right) n^{-\frac{2}{2+d}} .$$

*Proof.* Let $I$ denote the number of phases up to time $n$. We will decompose the expectation $\mathbb{E}_n$ in terms of the various phases $i \in [I]$ and apply Lemma 4. Let $B \triangleq \hat{C} \, 4^d \, \Delta_Y^2 + 12\lambda^2$. We have:

$$\mathbb{E}_{n} \mathbb{E}_{Y^n} |f_n(X) - f(X)|^2 \leq B \sum_{i=1}^{I} \frac{n_i}{n} n_i^{-\frac{2}{2+d}} = B \frac{I}{n} \sum_{i=1}^{I} \frac{1}{I} n_i^{\frac{d}{2+d}} \leq B \frac{I}{n} \left( \sum_{i=1}^{I} \frac{n_i}{I} \right)^{\frac{d}{2+d}}$$

$$= B \frac{I}{n} \left( \frac{n}{I} \right)^{\frac{d}{2+d}} = B \cdot I^{\frac{2}{2+d}} n^{-\frac{2}{2+d}} \leq B \cdot d^{\frac{2}{2+d}} n^{-\frac{2}{2+d}},$$

where in the second inequality we use Jensen's inequality, and in the last inequality Lemma 3. $\square$

## 4.2 Bound on $L_2$ Error

We need the following lemma, whose proof is in the supplemental appendix, which bounds the probability that a $\rho$-ball of a given radius contains a sample from $x^n$. This will then allow us to bound the bias induced by transforming a solution for the adversarial setting to a solution for the stochastic setting.

**Lemma 5.** *Suppose $(\mathcal{X}, \rho, \mu)$ has metric measure dimension $d$. Let $\mu$ be a distribution on $\mathcal{X}$ and let $\mu_n$ denote an empirical distribution on an i.i.d sample $x^n$ from $\mu$. For $\epsilon > 1/n$, let $\mathcal{B}_\epsilon$ denote the class of $\rho$-balls centered on $\mathcal{X}$ of radius $\epsilon$. There exists $C$ depending on $d$ such that the following holds. Let $0 < \delta < 1$, Define $\alpha_{n,\delta} = C \left( d \log n + \log(1/\delta) \right)$. Then, with probability at least $1 - \delta$, for all $B \in \mathcal{B}_\epsilon$ satisfying $\mu(B) \geq \alpha_{n,\delta}/n$ we have $\mu_n(B) > 1/n$.*

We are now ready to prove Theorem 1.

*Proof of Theorem 1.* Fix $\delta = 1/n$ and define $\alpha_{n,\delta}$ as in Lemma 5. Pick $\epsilon = (\alpha_{n,\delta}/C_1 n)^{1/d} \geq 1/n$, where $C_1$ is such that every $B \in \mathcal{B}_\epsilon$ has mass at least $C_1 \epsilon^d$. Since for every $B \in \mathcal{B}_\epsilon$, $\mu(B) \geq C_1 \epsilon^{-d} \geq \alpha_{n,\delta}/n$, we have by Lemma 5, that with probability at least $1 - \delta$, all $B \in \mathcal{B}_\epsilon$ contain a point from $x^n$. In other words, the event $\mathcal{E}$ that $x^n$ forms an $\epsilon$-cover of $\mathcal{X}$ is $(1 - \delta)$-likely.

Suppose $x_t$ is the closest point in $x^n$ to $x \in \mathcal{X}$. We write $x \to x_t$. Then, under $\mathcal{E}$, we have, $\|f(x) - f(x_t)\| \le \lambda\epsilon$. We therefore have by Fubini's theorem

$$
\begin{aligned}
\mathop{\mathbb{E}}_{x^n,Y^n} \|f_n - f\|_{2,\mu}^2 &= \mathop{\mathbb{E}}_{x^n} \mathop{\mathbb{E}}_{X} \mathop{\mathbb{E}}_{Y^n|x^n} |f_n(X) - f(X)|^2 \cdot (\mathbf{1}\{\mathcal{E}\} + \mathbf{1}\{\bar{\mathcal{E}}\}) \\
&\le \mathop{\mathbb{E}}_{x^n} \sum_{t=1}^{n} 2\mu(x : x \to x_t) \mathop{\mathbb{E}}_{Y^n|x^n} |f_n(x_t) - f(x_t)|^2 + 2\lambda^2\epsilon^2 + \delta\Delta_Y^2 \\
&\le \mathop{\mathbb{E}}_{x^n} \sum_{t=1}^{n} 2C_2\epsilon^d \mathop{\mathbb{E}}_{Y^n|x^n} |f_n(x_t) - f(x_t)|^2 + 2\lambda^2\epsilon^2 + \delta\Delta_Y^2 \\
&\le \frac{2C_2\alpha_{n,\delta}}{C_1} \sup_{x^n} \mathbb{E}_n \mathop{\mathbb{E}}_{Y^n} |f_n(x_t) - f(x_t)|^2 + 2\lambda^2\epsilon^2 + \delta\Delta_Y^2,
\end{aligned}
$$

where in the first inequality we break the integration over the Voronoi partition of $\mathcal{X}$ defined by the points in $x^n$, and introduce $f(x_t)$; the second inequality uses $\{x : x \to x_t\} \subset B(x_t, \epsilon)$ under $\mathcal{E}$. □

### 4.3 Lower-bound

Let's consider first the case of a fixed $n$. The idea behind the proof is as follows: for $\mu$ fixed, we have to come up with a class $\mathcal{F}$ of functions which vary considerably on the space $\mathcal{X}$. To this end we discretize $\mathcal{X}$ into as many cells as possible, and let any $f \in \mathcal{F}$ potentially change sign from one cell to the other. The larger the dimension $d$ the more we can discretize the space and the more complex $\mathcal{F}$, subject to a Lipschitz constraint. The problem of picking the right $f$ can thus be reduced to that of classification, since the learner has to discover the sign of $f$ on sufficiently many cells.

In order to handle many data sizes $n$ simultaneously, we borrow from the idea above. Say we want to show that the lower-bound holds for a subsequence $\{n_i\}$ simultaneously. Then we reserve a subset of the space $\mathcal{X}$ for each $n_1, n_2, \ldots$, and discretize each subset according to $n_i$. The functions in $\mathcal{F}$ have to then vary sufficiently in each subset of the space $\mathcal{X}$ according to the corresponding $n_i$. This is illustrated in Figure 3.

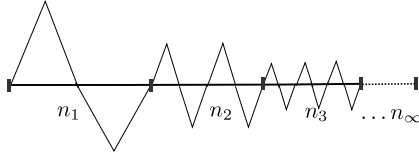

We can then apply the same idea of reduction to classification for each $n_t$ separately. This sort of idea appears in [7] where $\mu$ is uniform on the Euclidean cube, where they use the regularity of the cube to set up the right sequence of discretizations over subsets of the cube. The main technicality in our result is that we work with a general space without much regularity. The lack of regularity makes it unclear a priori how to *divide* such a space into subsets of the proper *size* for each $n_i$.

Figure 3: Lower bound proof idea.

Last, we have to ensure that the functions $f \in \mathcal{F}$ resulting from our discretization of a general metric space $\mathcal{X}$ are in fact Lipschitz. For this, we extend some of the ideas from [9] which handles the case of a fixed $n$. For lack of space, the complete proof is in the extended version of the paper.

## 5 Conclusions

We presented an efficient and nearly minimax optimal approach to nonparametric regression in a streaming setting. The streaming setting is gaining more attention as modern data sizes are getting larger, and as data is being acquired online in many applications.

The main insights behind the approach presented extend to other nonparametric methods, and are likely to extend to settings of a more adversrial nature. We left open the question of optimal adaptation to the smoothness of the unknown function, while we effciently solve the equally or more important question of adapting to the the unknown dimension of the data, which generally has a stronger effect on the convergence rate.

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
