[Supplementary Material]

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

\underset{x^n, Y^n}{\mathbb{E}} \|f_n - f\|_{2,\mu}^2 &= \underset{x^n}{\mathbb{E}} \underset{X}{\mathbb{E}} \underset{Y^n|x^n}{\mathbb{E}} |f_n(X) - f(X)|^2 \cdot (\mathbf{1}\{\mathcal{E}\} + \mathbf{1}\{\bar{\mathcal{E}}\}) \\
&\le \underset{x^n}{\mathbb{E}} \sum_{t=1}^{n} 2\mu(x : x \to x_t) \underset{Y^n|x^n}{\mathbb{E}} |f_n(x_t) - f(x_t)|^2 + 2\lambda^2\epsilon^2 + \delta\Delta_Y^2 \\
&\le \underset{x^n}{\mathbb{E}} \sum_{t=1}^{n} 2C_2\epsilon^d \underset{Y^n|x^n}{\mathbb{E}} |f_n(x_t) - f(x_t)|^2 + 2\lambda^2\epsilon^2 + \delta\Delta_Y^2 \\
&\le \frac{2C_2\alpha_{n,\delta}}{C_1} \underset{x^n}{\sup} \mathbb{E}_n \underset{Y^n}{\mathbb{E}} |f_n(x_t) - f(x_t)|^2 + 2\lambda^2\epsilon^2 + \delta\Delta_Y^2,
\end{aligned}
$$

where in the first inequality we break the integration over the Voronoi partition of $\mathcal{X}$ defined by the points in $x^n$, and introduce $f(x_t)$; the second inequality uses $\{x : x \to x_t\} \subset B(x_t, \epsilon)$ under $\mathcal{E}$. $\quad\square$

### 4.3 Lower-bound

Let's consider first the case of a fixed $n$. The idea behind the proof is as follows: for $\mu$ fixed, we have to come up with a class $\mathcal{F}$ of functions which vary considerably on the space $\mathcal{X}$. To this end we discretize $\mathcal{X}$ into as many cells as possible, and let any $f \in \mathcal{F}$ potentially change sign from one cell to the other. The larger the dimension $d$ the more we can discretize the space and the more complex $\mathcal{F}$, subject to a Lipschitz constraint. The problem of picking the right $f$ can thus be reduced to that of classification, since the learner has to discover the sign of $f$ on sufficiently many cells.

In order to handle many data sizes $n$ simultaneously, we borrow from the idea above. Say we want to show that the lower-bound holds for a subsequence $\{n_i\}$ simultaneously. Then we reserve a subset of the space $\mathcal{X}$ for each $n_1, n_2, \ldots$, and discretize each subset according to $n_i$. The functions in $\mathcal{F}$ have to then vary sufficiently in each subset of the space $\mathcal{X}$ according to the corresponding $n_i$. This is illustrated in Figure 3.

Figure 3: Lower bound proof idea.

We can then apply the same idea of reduction to classification for each $n_t$ separately. This sort of idea appears in [7] where $\mu$ is uniform on the Euclidean cube, where they use the regularity of the cube to set up the right sequence of discretizations over subsets of the cube. The main technicality in our result is that we work with a general space without much regularity. The lack of regularity makes it unclear a priori how to *divide* such a space into subsets of the proper *size* for each $n_i$.

Last, we have to ensure that the functions $f \in \mathcal{F}$ resulting from our discretization of a general metric space $\mathcal{X}$ are in fact Lipschitz. For this, we extend some of the ideas from [9] which handles the case of a fixed $n$. For lack of space, the complete proof is in the extended version of the paper.

## 5 Conclusions

We presented an efficient and nearly minimax optimal approach to nonparametric regression in a streaming setting. The streaming setting is gaining more attention as modern data sizes are getting larger, and as data is being acquired online in many applications.

The main insights behind the approach presented extend to other nonparametric methods, and are likely to extend to settings of a more adversrial nature. We left open the question of optimal adaptation to the smoothness of the unknown function, while we effciently solve the equally or more important question of adapting to the the unknown dimension of the data, which generally has a stronger effect on the convergence rate.

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

# 6 Appendix

## 6.1 Proof of Lemma 3

We require the following lemma that relates the size of a packing to that of a cover.

**Lemma 6** ([4]). *If $(\mathcal{X}, \rho)$ has metric dimension $d$ for some $C_\rho$ (see Definition 2), then any $\epsilon$-packing $Z$ of $\mathcal{X}$ (i.e. $\rho(z, z') > \epsilon$ for pairs $z, z' \in Z$) has size at most that $2^d C_\rho \epsilon^{-d}$.*

*Proof.* By the design of the algorithm, and the fact that we use a 2-approximate nearest neighbor search, we know that $\mathbf{X}_i$ is a $\frac{\epsilon_t}{2}$ packing, hence we can apply Lemma 6 to have $\mathbf{X}_i \leq 2^d C_\rho \frac{\epsilon_t}{2}^{-d}$, hence, by definition of $d_i$, we have $d_i \leq d$ whenever $\hat{C} \geq \hat{C}_\rho$. The lower-bound on $d_i$ is obtained by induction as follows. We have $d_1 = 1$. For $i > 1$, we have $d_{i+1} > d_i$ since it held that $|\mathbf{X}_i| + 1 > \hat{C} 4^{d_i} \epsilon_t^{-d_i}$. Thus $d_{i+1} \geq d_i + 1 \geq i + 1$.

For the second part, the bound holds by the condition on ending a phase and because $\epsilon_t$ decreases over time within a phase. $\square$

## 6.2 Proof of Lemma 5

*Proof.* By Lemma 1, there exists an $\epsilon/4$-cover $Z$ of $\mathcal{X}$ of size at most $(\epsilon/2)^{-d} \leq \hat{C}_0 (4n)^d$ for some $C_0$. Apply Bernstein's inequality for every $z \in Z$, followed by a union bound over $z \in Z$ to obtain that, with probability at least $1 - \delta$, $\mu_n(B(z, \epsilon/2)) > 1/n$ whenever $\mu(B(z, \epsilon/2)) \geq \alpha'_{n, \delta}/n$, where $\alpha'_{n, \delta} = C_1 (d \log n + \log(1/\delta))$ an appropriate setting of $C_1$. Now remark that every ball $B(x, \epsilon)$ in $\mathcal{B}_\epsilon$ contains a ball $B(z, \epsilon/2)$ which in turn contains $B(x, \epsilon/4)$. By assumption, there exist $C_2$ such that for all $x \in \mathcal{X}$, $C_2 \mu(B(x, \epsilon/4)) \geq \mu(B(x, \epsilon))$. Thus for $\mu(B(x, \epsilon)) \geq C_2 \alpha'_{n, \delta}/n$, we have $\mu(B(z, \epsilon/2)) \geq \mu(B(z, \epsilon/2)) \geq \alpha'_{n, \delta}/n$ and therefore $\mu_n(B(x, \epsilon)) \geq \mu_n(B(z, \epsilon/2)) \geq 1/n$. $\square$

## 6.3 Lower Bound Proof

We start with the following definition.

**Definition 4** ($\epsilon$-net). *An $\epsilon$-net is both an $\epsilon$-cover and an $\epsilon$-packing.*

**Theorem 3.** *Supppose $(\mathcal{X}, \mu, \rho)$ has diameter $1$, and measure-dimension $d$. Suppose $(\mathcal{X}, \rho)$ admits a metric measure $\mu$. For any $n \in \mathbb{N}$, define $r_n^2 = (\lambda^2 n)^{-\frac{2}{2+d}}$. Pick any positive sequence $\{\alpha_n\}_{n \in \mathbb{N}}$, $\alpha_n \xrightarrow{n \to \infty} 0$. There exists $C > 0$, and an indexing subsequence $\{n_t\}_{t \in \mathbb{N}}$, $n_{t+1} > n_t$, such that*

$$\inf_{\{f_n\}} \sup_{\mathcal{D}_\lambda} \overline{\lim_{t \to \infty}} \frac{\mathbb{E}_{X^{n_t}, Y^{n_t}} \|f_{n_t} - f\|_{2,\mu}^2}{\alpha_{n_t} \lambda^2 r_{n_t}^2} > C,$$

*where the infimum is taken over all sequences $\{f_n\}$ of estimators $f_n : X^n, Y^n \mapsto L_{2,\mu}$.*

*Proof.* For $t \in \mathbb{N}$, let $n_0 = 0$ and define $n_t$ recursively as the smallest value of $n > n_{t-1}$ such that $2^{-t} > \max \{8r_n, \alpha_n\}$.

Given $z \in \mathcal{X}$, and $t \in \mathbb{N}$, define the following $z$-centered function over $\mathcal{X}$:

$$g_{z,t}(x) \triangleq \frac{\lambda}{5} (\tau r_{n_t} - \rho(x, z))_+,$$

for some fixed $\tau \leq 1/2$ which is further specified below. In other words, $g_{z,n}(x) = 0$ whenever $\rho(x, z) \geq \tau r_{n_t}$. It is not hard to see that, by triangle-inequality, $g_{z,t}(x)$ is $\lambda/5$-Lipschitz.

We will choose centers $z$ and indices $t$ to form a collection $\mathcal{G} = \{g_{z,t}\}$ which generates a class of functions

$$\mathcal{F} = \left\{ f_\sigma(\cdot) = \sum_{g_{z,t} \in \mathcal{G}} \sigma_{z,t} \cdot g_{z,t}(\cdot), \sigma_{z,t} \in \{-1, 1\} \right\}.$$

The collection $\mathcal{G}$ is defined recursively as follows.

For $t \in \mathbb{N}$, let $Z_t$ denote a $2^{-t}$-net of $\mathcal{X}$. We will pick points in $Z'_t \subset Z_t$, and pick centers for functions $g_{z,t}$ near these points in $Z'_t$.

Let $Z'_0 = \emptyset$. For $t \geq 1$, let $\mathcal{B}_{t-1}$ denote $\cup_{s=0}^{t-1} \left\{ B(z, 2^{-s-1}) : z \in Z'_s \right\}$. Let $c_t = C_0 2^{t(d-1)}$ for $C_0 < C_1/(C_2+1)$. Pick $c_t$ points $z \in Z_t \setminus \mathcal{B}_{t-1}$. These are the points in $Z'_t$. This is always possible since

$$|Z_t \setminus \mathcal{B}_{t-1}| \geq |Z_t| - \sum_{s=0}^{t-1} \left| \left\{ B(z, 2^{-s-1}) \cap Z_t : z \in Z'_s \right\} \right|$$

$$\geq |Z_t| - \sum_{s=0}^{t-1} c_s C_2 2^{(t-s-1)d} = |Z_t| C_0 C_2 \sum_{s=0}^{t-1} 2^{(t-1)d-s}$$

$$\geq C_1 2^{td} - C_0 C_2 2^{(t-1)d} \geq C_0 2^{t(d-1)} = c_t.$$

Now, for every $z' \in Z'_t$, let $Z_{n_t}(z')$ be an $r_{n_t}$ packing of $B(z', 2^{-t}/8)$. We let $Z''_t = \cup_{z' \in Z'_t} Z_{n_t}(z')$. We can now define $\mathcal{G} \equiv \{g_{z,t} : z \in Z''_t, t \geq 1\}$. The functions in $\mathcal{G}$ are mutually orthogonal in $L_{2,\mu}$. We show this next.

Pick any $s \geq t \geq 1$, different centers $z_t \in Z''_t$, $z_s \in Z''_s$, and suppose $z_t \in Z_{n_t}(z'_t)$ and $z_s \in Z_{n_s}(z'_s)$ for some $z'_t \in Z'_t$ and $z'_s \in Z'_s$. If $z'_t = z'_s$, then $t = s$, and $\rho(z_t, z_s) > r_{n_t}$ by construction. Otherwise, if $z'_t \neq z'_s$, then

$$\rho(z_t, z_s) \geq \rho(z'_t, z'_s) - \rho(z'_t, z_t) - \rho(z'_s, z_s)$$
$$\geq 2^{-t}/2 - 2^{-t}/4 > r_{n_t}.$$

In either case, $g_{z_t,n_t}$ and $g_{z_s,n_s}$ are respectively nonzero only on balls $B(z_t, \tau r_{n_t})$ and $B(z_s, \tau r_{n,s})$, and these balls do not intersect since $\rho(z_t, z_s) > r_{n_t} \geq 2\tau r_{n_t}$. Thus $< g_{z_t,n_t}, g_{z_s,n_s} > = \mathbb{E}_{X \sim \mu} |g_{z_t,n_t}(X) \cdot g_{z_s,n_s}(X)| = 0$.

One can show that for $\tau$ sufficiently small (relative to the growth of $\mu$), the functions $f_\sigma \in \mathcal{F}$ are $\lambda$-Lipschitz. The argument, which can be found in Therorem 2 of [], relies on the fact that the functions $g_{z,t} \in \mathcal{G}$ are $\lambda/5$-Lipschitz, and that $\mu$ is doubling.

We will consider the class of distributions $\mathcal{D}_{\mathcal{F}}$ of $X, Y$, where $X \sim \mu$, and $Y = f_\sigma(X) + \mathcal{N}(0,1), f_\sigma \in \mathcal{F}$. By construction $\mathcal{D}_{\mathcal{F}} \subset \mathcal{D}_\lambda$ so we only need to bound the hardness of learning from $\mathcal{D}_{\mathcal{F}}$.

Now given the above construction, the rest of the proof relies on standard techniques (see e.g. [7]), where the regression problem is reduced to some classification problems.

We will need the following fact: there exists $C_3, C_4$ such that, for any $g_{z,t} \in \mathcal{G}$, we have

$$C_3 \lambda^2 r_{n_t}^{2+d} \leq \|g_{z,t}\|^2 \leq C_4 \lambda^2 r_{n_t}^{2+d}.$$

To see this, remark that $g_{z,t}$ is at most $\lambda \tau r_{n_t}$ on $B(z, \tau r_{n_t})$ and at least $\lambda \tau r_{n_t}/10$ on $B(z, \tau r_{n_t}/2)$; furthermore, by the doubling assumption on $\mu$, these balls have mass proportional to $r_{n_t}^d$.

Now, for every estimator $\{f_n\}$, we let $f_{n,\mathcal{G}} = \sum_{g_{z,t} \in \mathcal{G}} w_{z,t} g_{z,t}$ denote the projection of $f_n(X^n, Y^n)$ onto the orthonormal system induced by $\mathcal{G}$. We have for every $n_t$ that

$$\|f_{n_t} - f\|_{2,\mu}^2 \geq \|f_{n_t,\mathcal{G}} - f\|_{2,\mu}^2$$
$$\geq \sum_{z \in Z''_t} (w_{z,t} - \sigma_{z,t})^2 \|g_{z,t}\|^2$$
$$\geq \sum_{z \in Z''_t} \mathbf{1}\{w_{z,t} \cdot \sigma_{z,t} < 0\} \|g_{z,t}\|^2$$
$$\geq C_3 \lambda^2 r_{n_t}^{2+d} \sum_{z \in Z''_t} \mathbf{1}\{w_{z,t} \cdot \sigma_{z,t} < 0\}.$$

Let $\mathbb{E}_\sigma$ denote expectation over the random choice of $\sigma = \{\sigma_{z,t}\}$ where each $\sigma_{z,t} = 1$ w.p. $1/2$. Let $\mathbb{E}$ denote $\mathbb{E}_{X^{n_t}, Y^{n_t}}$ for short. We have

$$
\inf_{\{f_n\}} \sup_{\mathcal{D}_{\mathcal{F}}} \overline{\lim_{t \to \infty}} \frac{\mathbb{E} \|f_{n_t} - f\|_{2,\mu}^2}{\alpha_{n_t} \lambda^2 r_{n_t}^2}
$$

$$
\geq \inf_{\{f_n\}} \mathbb{E}_\sigma \overline{\lim_{t \to \infty}} \frac{\mathbb{E} \|f_{n_t} - f\|_{2,\mu}^2}{\alpha_{n_t} \lambda^2 r_{n_t}^2}
$$

$$
\geq \inf_{\{f_n\}} \mathbb{E}_\sigma \overline{\lim_{t \to \infty}} \frac{C_3 r_{n_t}^d \, \mathbb{E} \sum_{z \in Z_t''} \mathbf{1}\{w_{z,t} \cdot \sigma_{z,t} < 0\}}{\alpha_{n_t}}
$$

$$
\geq \inf_{\{f_n\}} \mathbb{E}_\sigma \overline{\lim_{t \to \infty}} \frac{C_3 r_{n_t}^d \sum_{z \in Z_t''} \mathbb{E}\, \mathbf{1}\{w_{z,t} \cdot \sigma_{z,t} < 0\}}{2^{-t}}
$$

$$
\geq \inf_{\{f_n\}} \overline{\lim_{t \to \infty}} \frac{C_3 r_{n_t}^d \sum_{z \in Z_t''} \mathbb{E}\,\mathbb{E}_\sigma \mathbf{1}\{w_{z,t} \cdot \sigma_{z,t} < 0\}}{2^{-t}},
$$

where in the last inequality above we applied, first, the dominated convergence theorem (DCT), then Fubini's theorem. The application of DCT is valid since the term in the limit can be upper-bounded by noticing that

$$
r_{n_t}^d \sum_{z \in Z_t''} \mathbb{E}\,\mathbf{1}\{w_{z,t} \cdot \sigma_{z,t} < 0\}
$$

$$
\leq r_{n_t}^d |Z''t| \leq r_{n_t}^d c_t C_2 2^{-(t+3)d} r_{n_t}^{-d} = C_0 C_2 2^{-t+3d} \ .
$$

For $z \in Z_t'$, and fix $X^{n_t}$. Then $\mathbb{E}\,\mathbb{E}_\sigma \mathbf{1}\{w_{z,t} \cdot \sigma_{z,t} < 0\}$ is the probability of error of a classifier (which outputs $\mathrm{sign}(w_{n,z})$) for the following prediction task. Let $x_{(1)}, x_{(2)}, \dots x_{(m)}$ denote the values of $X$ falling in $B(z, \tau r_{n_t})$ where $g_{z,t}$ is non zero. Then

$$
(Y_{(1)}, \dots Y_{(m)}) = \varsigma_z(g_{z,t}(x_{(1)}), \dots, g_{z,t}(x_{(m)})) + \mathcal{N}(0, I_m)
$$

is a random vector sampled from the equal-weight mixture of two spherical Gaussians in $\mathbb{R}^m$ centered at $u \doteq (g_{z,t}(x_{(1)}), \dots, g_{z,t}(x_{(m)}))$ and $-u$. The prediction task is that of identifying the right mixture component from the single sample $(Y_{(1)}, \dots Y_{(m)})$. The smallest possible error for this task is that of the Bayes classifier and is well known to be $\Phi(-\|u\|) \geq \Phi\left(-\sqrt{\sum_{i=1}^{n_t} g_{z,t}^2(X_i)}\right)$. We can now take the expectation over $X^{n_t}$, and since $\Phi(-\sqrt{\cdot})$ is convex, we have by Jensen's inequality that.

$$
\mathbb{E}\,\mathbb{E}_\sigma \mathbf{1}\{w_{z,t} \cdot \sigma_{z,t} < 0\} \geq \mathbb{E}_{X^{n_t}} \Phi\left(-\sqrt{\sum_{i=1}^{n} g_{z,t}^2(X_i)}\right)
$$

$$
\geq \Phi\left(-\sqrt{\sum_{i=1}^{n} \mathbb{E}_{X_i} g_{z,t}^2(X_i)}\right) = \Phi\left(-\sqrt{n \|g_{z,t}\|^2}\right)
$$

$$
\geq \Phi\left(\sqrt{C_4}\right) \ .
$$

Finally, since $|Z_t''| \geq c_t C_1 2^{-(t+3)d} r_{n_t}^{-d} = C_0 C_1 2^{-t+3d} r_{n_t}^{-d}$, we have

$$
\inf_{\{f_n\}} \sup_{\mathcal{D}_{\mathcal{F}}} \overline{\lim_{t \to \infty}} \frac{\mathbb{E} \|f_{n_t} - f\|_{2,\mu}^2}{\alpha_{n_t} \lambda^2 r_{n_t}^2} \geq C_3 C_0 C_1 2^{3d} \Phi\left(\sqrt{C_4}\right).
$$

$\square$

The main lower bound statement of Theorem 2 is obtained as follows.

*Proof of Theorem 2.* Fix any $f_n$. Let $\alpha_n^2 = \beta_n/(\lambda^2 r_n^2)$. From Theorem 3 there exists a distribution in $\mathcal{D}_\lambda$ such that

$$
\overline{\lim_{n \to \infty}} \frac{\mathbb{E}_{X^n, Y^n} \|f_{n_t} - f\|_{2,\mu}^2}{\alpha_n^2 \lambda^2 r_n^2} = \infty.
$$

Pick a subsequence whose limit is the lim sup. $\square$