[Reviews · NeurIPS 2013]

Submitted by Assigned_Reviewer_2

The authors derive a new procedure to estimate a recursive-partition
prediction rule in the streaming framework. Theoretical analyses
demonstrate that the procedure is computationally efficient and attains the
minimax prediction error rate, up to a log factor. A small empirical
analysis is in agreement with the theory.

The paper is excellent: the authors have produced an intuitive streaming
algorithm with nearly sharp theoretical guarantees in terms of intrinsic
dimension. The exposition is very clear and easy to follow. I have been
hoping for some time to see a streaming regression tree paper like this.
Properly publicized, this work is liable to have a big impact in the
large-scale learning community. Regarding originality, the authors have
overlooked some work on this problem -- but that work focuses on bounding
the difference between the streaming tree and the corresponding batch tree,
which is a different question altogether. So there's no harm done. I've
listed some references below.

I warmly recommend acceptance.

Minor comments.

p.2: "everywhere on the space rho" -> "everywhere on the space calX"
p.2: "The metric (calX, rho)" -> "The metric space (calX, rho)"
p.3: "an new guess" -> "a new guess"
p.3: "quantifieable" -> "quantifiable" [***** SPELL CHECK *****]
p.3: "there exist C" -> "there exists C"

References.

@inproceedings{,
author = {Domingos, Pedro and Hulten, Geoff},
booktitle = {Proceedings of the 6th ACM SIGKDD International Conference on Knowledge Discovery and Data Mining},
title = {Mining high-speed data streams},
pages = {71--80},
year = {2000}
}

@inproceedings{,
author = {Hulten, Geoff and Spencer, Laurie and Domingos, Pedro},
booktitle = {Proceedings of the 7th ACM SIGKDD International Conference on Knowledge Discovery and Data Mining},
title = {Mining time-changing data streams},
pages = {97--106},
year = {2001}
}

@inproceedings{,
author = {Pfahringer, Bernhard and Holmes, Geoffrey and Kirkby, Richard},
booktitle = {Advances in Knowledge Discovery and Data Mining: Proceedings of the 12th Pacific-Asia Conference (PAKDD)},
title = {Handling numeric attributes in Hoeffding trees},
volume = {5012},
pages = {296--307},
publisher = {Springer},
year = {2008}
}

@article{,
author = {Ben-Haim, Yael and Tom-Tov, Elad},
title = {A streaming parallel decision tree algorithm},
journal = {Journal of Machine Learning Research},
volume = {11},
pages = {849--872},
year = {2010}
}

NB: I have read the author rebuttal.
Summary: Accept.

Submitted by Assigned_Reviewer_5

The paper proposes an algorithm for nonparametric learning in a streaming scenario where data is received i.i.d. over time. The proposed algorithm is an incremental tree-based regression algorithm where clusters of connected points are grown as the data is received and assigned the same prediction. The main contributions of the paper are: (1) A rule that determines when to start a new tree and when to continue growing an existing tree. It is based on an estimate of current problem's dimensionality. (2) An analysis of complexity and convergence proofs for the proposed rule, showing that the proposed rule has advantageous asymptotic behavior.

My background in formal learning theory is limited. My understanding of the paper is shallow and my review does not include a serious investigation of the proofs. However, I have the general impression that the work is well-conducted and practically relevant.

The paper is well written, as illustrated by the numerous high-level explanations that accompany the technical content. This makes the paper readable, even by a non-specialist.

The authors are testing their method on two datasets (an artificial one and a real one) showing that a dynamic estimation of the dimensionality competes with the optimal dimensionality parameter which is a priori unknown.

As a limitation of the current work, only a global measure of dimensionality over the whole input space is considered. In many practical applications, a local dimensionality estimate would be more appropriate.

Some typos:

l.060: proveably -> provably
l.139: an new guess -> a new guess
l.148: quantifieable -> quantifiable
l.191: usually is usually -> usually
l.428: effciently -> efficiently
Summary: From what I could understand, I think that the work is well-conducted and practically relevant. I appreciate that the authors made the paper readable to a certain degree by a non-specialist.

Submitted by Assigned_Reviewer_6

This paper is well written and its theoretical results appear interesting and strong, certainly of interest to the wider field.

On the other hand, the authors' claim to methodological originality is not well founded, and should be revised for subsequent versions of the paper. They say "We know of no other work concerning the problem of maintaining a tree-based regressor in a streaming setting." They can't have searched very hard, as they missed two recent papers in flagship journals, both of which turned up for me with a simple web search for "dynamic tree model":

1) Variable selection and sensitivity analysis using dynamic trees, with an application to computer code performance tuning. Robert B. Gramacy, Matt Taddy, and Stefan M. Wild. Annals of Applied Statistics Volume 7, Number 1 (2013), 51-80.

2) Dynamic Trees for Learning and Design. Matthew A. Taddy, Robert B. Gramacy, Nicholas G. Polson. Journal of the American Statistical Association, Volume 106, Issue 493, 2011.

I am certainly not implying that these papers deprecate the authors' theoretical findings -- I haven't read them closely, but they certainly don't appear to have similar theoretical results. But these are major venues (JASA, AoAS), and there's an R package for one of these papers that would presumably make for easy benchmarking (http://www.cran.r-project.org/web/packages/dynaTree/index.html).
Summary: More homework needed.
Author Feedback

Author rebuttal: We thank the Reviewers for their precise and useful comments.
We especially thank them for pointing out relevant references we were unaware of.

We would like to stress however, as already noticed by the reviewers, that these other references do not consider the non-trivial theoretical question addressed in our paper, namely the possibility of updating a tree-based regressor in a streaming setting under the constraints of O(log n) update and nearly minimax-optimal rates in terms of the (unknown) intrinsic dimension. Our general algorithmic approach turned out simple and novel, and our theory is easily validated through simulations.

Nonetheless, the pointed-out references are relevant and the paper will be appropriately updated.